# The Chemistry of Green and Roasted Coffee by Selectable 1D/2D Gas Chromatography Mass Spectrometry with Spectral Deconvolution

**DOI:** 10.3390/molecules27165328

**Published:** 2022-08-21

**Authors:** Scott C. Frost, Paige Walker, Colin M. Orians, Albert Robbat

**Affiliations:** 1Department of Biology, Tufts University, Medford, MA 02155, USA; 2Department of Chemistry, Tufts University, Medford, MA 02155, USA

**Keywords:** GC/MS, 2D–GC/MS, deconvolution, coffee, green coffee, coffee quality

## Abstract

Gas chromatography/mass spectrometry (GC/MS) is a long-standing technique for the analysis of volatile organic compounds (VOCs). When coupled with the Ion Analytics software, GC/MS provides unmatched selectivity in the analysis of complex mixtures and it reduces the reliance on high-resolution chromatography to obtain clean mass spectra. Here, we present an application of spectral deconvolution, with mass spectral subtraction, to identify a wide array of VOCs in green and roasted coffees. Automated sequential, two-dimensional GC-GC/MS of a roasted coffee sample produced the retention index and spectrum of 750 compounds. These initial analytes served as targets for subsequent coffee analysis by GC/MS. The workflow resulted in the quantitation of 511 compounds detected in two different green and roasted coffees. Of these, over 100 compounds serve as candidate differentiators of coffee quality, AAA vs. AA, as designated by the Coopedota cooperative in Costa Rica. Of these, 72 compounds survive the roasting process and can be used to discriminate green coffee quality after roasting.

## 1. Introduction

Gas chromatography/mass spectrometry (GC/MS) is the most accurate, precise, selective, and sensitive means of analyzing volatile and semi-volatile organic compounds (VOCs). Some analysts operate the MS in selected ion monitoring mode to increase sensitivity, but in doing so selectivity is compromised, especially when 1 or 2 ions are used to identify target compounds [1]; the technique cannot be used to analyze for unknowns. In contrast, 2-dimensional GC/MS (GC-GC or GCxGC) provides unmatched increases in separation space, often resulting in clean spectra for both target compounds and unknowns. For complex mixture analysis, combining 2D GC/MS with spectral deconvolution software improves the analytical metrics on four benchmarks: accuracy, precision, selectivity, and sensitivity. In 2017, we reviewed the performance of spectral deconvolution algorithms to identify VOCs in complex samples analyzed by GC/MS [2]. The aim of deconvolution algorithms is to untangle fragmentation patterns of coeluting compounds, followed by identification using reference libraries, such as NIST, Wiley, Adams, etc. Our review included instrument-specific software (e.g., ChromaTOF (LECO), MassHunter Profinder (Agilent), and MassLynx (Waters), as well as software that work on a wider range of data file formats (e.g., BinBase, ADAP-GC 2.0, AutoDecon, MetaboliteDetector, MetaboAnalyst, MetabolomeExpress Project, MetAlign, mMass, MZmine, OpenChrom, PyMS, PYQAN, SpectConnect, TagFinder, AMDIS, and Ion Analytics). Although all of these tools are able to match sample spectra with library spectra, only BinBase, MassHunter Profiler, and Ion Analytics (IA) offer the ability to annotate databases, compare outputs across samples, and track each compound’s concentration across datasets.

In contrast to the BinBase and Mass Profiler software that rely on high resolution MS instruments, the Ion Analytics (IA) software processes both high- and low-resolution data, as well as GC-GC/MS [3,4,5,6,7] and GCxGC/MS data [8,9] (see, also, algorithmic approaches for handling GCxGC/MS data [10]). IA is different from other software; it also subtracts each target compound’s spectrum from the total ion current (TIC) peak. Then, it determines if additional characteristic fragmentation patterns exist. If so, and if the spectra across the peak are constant, reference libraries are used to identify the compound. If not, processing continues until the remaining spectra are subtracted and ion signals approximate background signals. The workflow provides the means to identify unknowns from knowns [2,11,12,13].

Recently, GC/MS [14], GCxGC/MS [10], commercial, and freely available algorithms and software were reviewed [15,16,17]. Although AMDIS, from the National Institute of Standards and Technology (http://chemdata.nist.gov/mass-spc/amdis/overview.html, accessed on 18 July 2022), continues to be the most often used deconvolution tool (free to end-users and vendors alike), growing interest exists in developing new algorithms and software to overcome its limitations. These include Bayesian [18] and multivariate statistics (with/without AMDIS) [19,20,21], as well as band-target entropy minimization [22,23,24], evolving window factor analysis-multivariate curve resolution [25], PARAFAC2 [26,27], autoGCMSDataAnal (with/without AMDIS) [28,29], MS-DIAL [30,31], and machine learning tool algorithms [32,33,34,35]. Toward this end, we continue to improve the operational aspects of both the workflow and Ion Analytics software to assess differences in tea [11], essential oils [36], gin [4], and other natural products.

Here, we apply GC-GC/MS and GC/MS with Ion Analytics spectral deconvolution workflow to better understand the volatile and semi-volatile profiles of two green coffees and their roasted counterparts. Green and roasted coffees are characterized by a wide array of volatile organics, responsible for the complexity and richness of the brewed coffee aroma [37]. Their identity is necessary to understand coffee quality and flavor. Our objective is to describe the chemistry of green and roasted coffees, while demonstrating the effectiveness of the Ion Analytics workflow to do so. Toward this end, we found 72 compounds that survive the roasting process and serve as potential markers of high-quality coffee.

## 2. Results and Discussion

### 2.1. Coffee Sourcing and Description

Green coffee was sourced from the Coopedota cooperative in the Tarrazú region of Costa Rica. The Tarrazú region is globally recognized for producing premium specialty coffee grown from *Coffea arabica* plants. The Coopedota cooperative sells processed and dried, green coffee for export with seven different quality designations. The two top premium designations are AAA and AA. The cooperative determines quality by measuring the ripeness of the freshly harvested coffee cherry upon its arrival to the cooperative for processing. Coffee cherries that meet the highest standard are designated as AAA and are processed separately from the AA cherry. The remaining five quality designations are derived from the AA coffee based on size and shape of the coffee post-processing; for example, chipped and broken green coffee is separated as low quality. Two of the green coffees and their roasted counterparts were used for this study, viz., the higher quality AAA coffee and lower quality AA coffee. We refer to these four samples as Green AAA, Green AA, Roasted AAA, and Roasted AA.

### 2.2. Building the Initial Target Compound Library

All four samples were subjected to the Ion Analytics workflow, with the results included in the final compound library. For example, Figure 1 displays the 1D chromatogram of the Roasted AAA extract on both the polar and non-polar columns. In contrast, Figure 2 depicts the automated sequential, GC-GC/MS chromatograms of two 1-min sample portions transferred from the polyethylene (WAX) column to the diphenyl dimethyl polysiloxane (RXI-5MS) column at minutes 20–21 and 30–31. As opposed to the few peaks shown in the same 1-min retention windows on the 1D wax column, a total of 37 and 50 compounds were found in the 2D sample fractions, respectively. As stated earlier, GC-GC increases separation efficiency compared to GC and provides the means to obtain clean spectra; it also increases the cost and requires longer analysis time. The IA workflow begins by inspecting each peak in the first separation (the 0–1 min heartcut) to determine if coelution exits. If not, sample spectra are compared to commercial library spectra to tentatively assign compound identity based on the quality control criterion described in the experimental section. When reference standards are available, compound identities are confirmed by comparing retention times and spectra. Then, the IA software automatically inputs the name of the compound, CAS #, retention time (index), mass spectrum (used to subtract fragmentation patterns), 3–5 target ions and their relative abundance (used to deconvolve target compounds), and any other information of importance (e.g., sample type, sensory response) into the database.

For each heartcut, the preceding analysis and, thus, the cumulative library-building process was used to identify organics in the AAA coffees. This is an important step in the workflow due to transference of sample components from the first to the second column as a result of flow conditions and concentration. This finding is not surprising given the number of compounds and coelutions shown in the two examples, see Figure 2. As stated earlier, the software inspects each peak scan, finds 3–5 invariant scans, and subtracts the average spectrum from the total ion current (TIC) peak to reveal the fragmentation pattern of the unresolved compound(s). The process continues until residual signals approximate background signals. This workflow produces “clean” compound spectra that are compared to library spectra, assigned tentative identities, and confirmed where possible at a later time. If sample and library spectra cannot be matched, a numerical identifier is assigned, with all of the analytic information related to that compound uploaded to the database.

In sum, nearly 1000 compounds were detected in the AAA coffee database. This extraordinary number, due mostly to the roasted coffee, was possible because we overloaded the first column knowing that the 1-min sample portion would be sufficiently separated on the second column. The same spectral deconvolution workflow was used to analyze the AA coffees by GC/MS; in this case, AAA compounds are the target compounds.

### 2.3. Applying the Coffee Library

First the AAA coffees were reanalyzed using GC/MS to determine if the deconvolution and MS subtraction algorithms correctly identified the target compounds. Then, the AA coffees were analyzed by GC/MS to identify differences between the two quality samples, thereby increasing the number of targets in the coffee library. For these analyses, the on-column mass was intentionally lowered to avoid overloading the non-polar column. A total of 511 compounds were identified among the four coffee samples, which is significantly less than that obtained by GC-GC/MS. Although excellent for library-building purposes, GC-GC/MS cannot be used to measure quantitative differences in samples due to sample carryover from one heartcut to the next.

Figure 3A,B show the GC/MS deconvolved, reconstructed ion current chromatograms, and the corresponding background signals (red line) after each compound’s fragmentation pattern was subtracted. The figure legend lists the post-deconvolution identity of each peak and the heartcut from which the AAA spectra were initially collected. In Figure 3A, compound identification for 426 and 437 was straightforward compared to 440, 445, and 448. After deconvolution, the resulting spectra were compared to those in the database. Although the 440 and 448 compounds elute at the same time with similar peak heights and are dominated by 445, all three were correctly identified. Once the spectra for these compounds were subtracted from the chromatogram, the residual ions approximated the background signal (red line). Figure 3B shows the challenge of identifying seven compounds that elute within ~0.15 s of each other. It is worth noting that 897 and 901 both separate in 1D and 2D GC/MS (29–30 min heartcut). To confirm the identity of 924, 925, and 926, the spectra of the other compounds were subtracted. Then, the software inspected each peak scan, identified 3–5 constant scans, and compared the average spectra to database spectra to determine compound identity. Finally, all seven spectra were subtracted from the corresponding peaks to reveal the baseline response in red. This approach provides the means to systematically identify all sample components by GC/MS. As the database grows, the software simplifies the analysis by subtracting target compound spectra from the chromatogram, yielding unknowns without, for the most part, coeluting ion signals. As each sample is analyzed the database is annotated increasing the number of targets for each succeeding sample analyzed. The detection of new compounds, therefore, are caused by differences in local growing (green coffee) and roasting conditions.

### 2.4. Comparison of Green and Roasted Coffees

Identifiable compounds were not equally distributed among the samples. Figure 4 displays a four-way Venn diagram illustrating both the differences and commonality of compounds among each of the four coffee samples. The figure is organized as four overlapping ovals, one for each of the four coffee samples. Each overlap is labeled with the number of compounds corresponding to that overlap. For example, eight compounds were found in the Green AA, Green AAA, and Roasted AAA, but not in the Roasted AA coffee. This is indicated by the number 8 in the overlapping portion of the three ovals for Green AA, Green AAA, and Roasted AAA. Although Green AA and Green AAA have 112 common compounds, 47 additional compounds were detected in Green AAA that were not expressed in Green AA. Of these 47 compounds identified in the Green AAA, 24 were unique to Green AAA, 9 were in common with the Roasted AAA, 13 were in common with the Roasted AAA and Roasted AA, and 1 was in common with only Roasted AA (Figure 4).

### 2.5. Comparison of Green Coffees

Of the 20 common Green AA and Green AAA compounds (Figure 4), 6 showed a two-fold or greater difference in the Green AAA (higher quality) to Green AA (lower quality) concentration ratio. These compounds were 2,5-dimethyl-3-(2-methylbutyl)pyrazine (8.4-fold), ethyl 4-ethoxybenzoate (2.9-fold), unknown 93 (2.9-fold), phenyl ethyl alcohol (2.8-fold), 2-methylindoline (2.5-fold), and benzyl benzoate (2.0-fold). In contrast, four compounds had a two-fold or greater Green AA to Green AAA concentration ratio, namely, trans methyl dihydrojasmonate (17.7-fold), 1-dodecanol (8.3-fold), unknown 366 (2.3-fold), and (*Z*)-2-methyl-6-(1-propenyl) pyrizine (2.0-fold).

Of the 112 shared Green AA and Green AAA compounds, 10 had a difference in concentration of 10-fold or higher after calculating the Green AAA to Green AA ratio. These compounds were hexadecenoic acid (179-fold), *n*-nonanal (43-fold), and pyridine (38-fold), 2-methyl-1-(1,1-dimethylethyl)-2-methyl-1,3-propanediyl ester (27-fold), 2-ethyl-6-methyl-pyrizine (23-fold), benzophenone (19-fold), 2-ethyl-5-methyl-pyrizine (14-fold), 5-methyl-furfural (12-fold), and 2,6,-diethyl-3-pyrizine (10-fold), whereas only four compounds were 10-fold or higher in the Green AA compared to Green AAA concentration ratio. These included trans methyl dihydrojasmonate (18-fold), 2,3-dihydro-1*H*-indole-1-carbaldehyde (13-fold), 2-ethylpyrazine (13-fold), and 2-phenylbut-2-enal (12-fold). This set of 14 compounds serves as a distinguishing indicator of the two coffee quality levels, the higher quality Green AAA and the lower quality Green AA.

The Green AAA coffee was further distinguished by the presence of 24 unique compounds (Figure 4; Table 1). Two highly notable sensory-active compounds were among the 24, nerol oxide, and cis–linalool oxide (floral and fruity aromas). Additionally, 2,6-dimethylpyrizine, 3-ethyl-2,5-dimethylpyrazine, and 2-phenethylacetate were not detected in either the Green AA or Roasted AA coffees but were measurable in the Roasted AAA coffee, thus, providing another chemical quality difference indicator. Lastly, one compound was identified as an unknown in the Green AA coffee (Figure 4, Table 1). The 25 compounds listed in Table 1 are candidates for further work when evaluating green coffee quality differences.

**Table 1 molecules-27-05328-t001:** Compounds unique to Green AAA and Green AA, their relative abundance, retention time, and odor characteristics.

Compound	Green AAA	Green AA	Roast AAA	Roast AA	Retention Time	Lit Odor
** *Unique to Green AAA* **						
171	40,806	0	0	0	19.05	nr
3-methyl-1-butanol	19,745	0	0	0	4.28	fermented
3-hexen-2-one	16,409	0	0	0	6.80	nr
methyl benzoate	12,172	0	0	0	14.23	phenolic
2-methyl-cyclopentanone	12,079	0	0	0	6.80	nr
2-ethyl furan	11,877	0	0	0	3.78	chemical
2,4-dimethyl-1,3-pentadiene,	10,381	0	0	0	3.78	caramel
methyl 3-(3,5-di-tert-butyl-4-hydroxyphenyl)propionate	10,130	0	0	0	36.71	nr
643	9657	0	0	0	28.22	
6-methyl-5-hepten-2-one	9540	0	0	0	10.70	citrus, green
217	9156	0	0	0	17.38	
nerol oxide	8261	0	0	0	16.57	green
175	8223	0	0	0	13.73	
3,5-dimethyl-2-isobutylpyrazine	8083	0	0	0	17.58	cocoa
265	7659	0	0	0	19.85	
431	7345	0	0	0	25.35	
4-methylpyrrolo [1,2-*a*]pyrazine	7323	0	0	0	21.47	nr
1-phenyl-2-butanone	7244	0	0	0	18.35	earthy
cyclopentanone	7172	0	0	0	5.26	nr
457	6984	0	0	0	26.68	
568	6590	0	0	0	31.47	
*cis*-linalool oxide (furan)	5701	0	0	0	13.51	earthy, floral
46	5528	0	0	0	10.59	
ortho-hydroxybiphenyl	5516	0	0	0	26.52	nr
** *Unique to Green AA* **						
450	0	24,258	0	0	12.00	

### 2.6. Comparison of Roasted AAA to Roasted AA

A total of 209 compounds were found in the Roasted AAA and AA coffees that were not detected in the corresponding green coffee samples. From this set, 11 compounds in the Roasted AAA sample were at least two-fold or higher in concentration compared to the AA sample. Four of them were 5-fold or higher in concentration: benzaldehyde (16.3-fold), 1-(2,4-dihydroxyphenyl)-ethanone (10.6-fold), benzophenone (7.3-fold), and 1-(2-hydroxyphenyl)propan-1-one (6.1-fold). Surprisingly, the remaining compounds, 198 of them in the Roasted AA coffee, were higher in concentration. Eight compounds were more than 5-fold higher, these included dimethyltrisulfide (31.1-fold), E-2-nonen-1-al (24.0-fold), 4-vinyl-2-methoxyphenol (12.1-fold), methyl 3-methylbutanoate (8.5-fold), unknown 641 (7.1-fold), 1-methyl-2,3-dihydroindole-5-carbaldehyde (7.1-fold), hexanal, (5.7-fold), and 1-(1*H*-pyrrol-2-yl)ethanone (5.6-fold). Dimethyltrisulfide (DMTS) and benzaldehyde exhibited the greatest disparity between the two quality levels. Although benzaldehyde is known to exhibit a nutty almond aroma, prior work showed its presence was consistent with lower quality green coffee [38]. In addition to the 209 compounds found in common between the Roasted AAA and Roasted AA, 74 compounds were unique to Roasted AAA, and 56 were unique to the Roasted AA (Table 2).

**Table 2 molecules-27-05328-t002:** Compounds unique to Roasted AAA and Roasted AA.

Compound	Green AAA	Green AA	Roast AAA	Roast AA	Retention Time	Lit Odor
** *Unique to Roast AAA* **						
4-(2-methylprop-2-enoyloxy)butyl 2-methylprop-2-enoate	0	0	1,872,381	0	27.19	nr
2,3,5-trimethylpyrazine	0	0	795,151	0	11.16	nutty
1-[1-(furan-2-ylmethyl)pyrrol-2-yl]ethanone	0	0	694,986	0	25.52	nr
furan-2-ylmethyl pentanoate	0	0	194,721	0	18.19	nr
719	0	0	191,413	0	13.26	
3-methylpentane-2,4-dione	0	0	182,840	0	14.56	nr
5-ethyl-2,3-dimethylpyrazine	0	0	142,250	0	13.98	burnt
(*E*)-3-(furan-2-yl)-2-methylprop-2-enal	0	0	128,245	0	17.29	spicy
678	0	0	123,536	0	31.43	
3-phenyl-2-propenal	0	0	87,809	0	19.47	spicy
694	0	0	82,049	0	33.61	
1-(4-methoxyphenyl)-propan-1-one	0	0	80,471	0	23.51	musty
pentane-2,3-dione	0	0	74,227	0	3.70	buttery
2,3-dihydro-1-benzofuran	0	0	73,915	0	18.19	phenolic
472	0	0	68,503	0	24.85	
503	0	0	67,741	0	24.62	
99	0	0	64,820	0	9.71	
473	0	0	64,012	0	25.16	
732	0	0	62,312	0	17.30	
406	0	0	57,488	0	23.40	
390	0	0	52,419	0	19.96	
305	0	0	48,447	0	19.08	
536	0	0	46,637	0	26.24	
344	0	0	42,780	0	20.71	
2-pyridinemethanol	0	0	42,327	0	20.18	nr
2-furanacetaldehyde-α-propyl	0	0	37,037	0	16.58	nr
furfuryl formate	0	0	36,063	0	8.24	nr
252	0	0	36,031	0	17.51	
321	0	0	34,425	0	18.49	
terpineol	0	0	33,957	0	17.28	citrus, woody
2-acetyl-4-methylpyridine	0	0	33,281	0	14.61	nr
427	0	0	31,760	0	24.89	
395	0	0	29,745	0	24.01	
417	0	0	28,972	0	25.79	
hexane-2,3-dione	0	0	27,926	0	5.20	buttery
118	0	0	27,593	0	16.23	
4-ethenyl-2,6-dimethoxyphenol	0	0	27,338	0	27.80	burnt
2-ethoxyaniline	0	0	26,009	0	16.49	nr
616	0	0	24,918	0	30.13	
2-acetyl-3-ethylpyrizine	0	0	24,790	0	16.52	nutty
579	0	0	24,744	0	25.62	
258	0	0	24,703	0	20.21	
2-methylthiolan-3-one	0	0	24,697	0	10.73	sulfurous
2-cyclohexene-1,4-dione	0	0	22,742	0	12.13	nr
73	0	0	21,729	0	8.29	
652	0	0	20,165	0	28.20	
3-methylbut-2-enyl acetate	0	0	19,106	0	8.77	fruity
218	0	0	18,875	0	17.43	
674	0	0	18,580	0	29.94	
333	0	0	18,410	0	21.44	
334	0	0	17,704	0	21.51	
313	0	0	17,266	0	20.93	
206	0	0	16,915	0	15.48	
691	0	0	16,395	0	32.39	
(*E*)-but-2-enal	0	0	16,165	0	4.43	nr
371	0	0	15,208	0	20.35	
436	0	0	15,106	0	21.32	
131	0	0	14,928	0	14.71	
1-phenyl-propan-1-one	0	0	14,205	0	16.48	floral, fruity
250	0	0	14,062	0	16.47	
730	0	0	13,064	0	15.32	
214	0	0	12,678	0	17.24	
204	0	0	12,331	0	15.32	
3-methylthiophene	0	0	11,185	0	5.04	fatty
48	0	0	10,644	0	12.99	
351	0	0	10,465	0	21.54	
40	0	0	9174	0	6.77	
6-tridecyloxan-2-one	0	0	8075	0	40.72	fatty
37	0	0	7152	0	9.54	
3-methylbut-2-en-1-ol	0	0	6839	0	5.07	sweet fruit
1,3,5-trimethylbenzene	0	0	6117	0	10.88	nr
3-methylbut-3-en-1-ol	0	0	5895	0	4.26	nr
27	0	0	5870	0	9.20	
725	0	0	5115	0	6.82	
** *Unique to Roast AA* **						
1-(furan-2-ylmethyl)pyrrole	0	0	0	938,004	16.40	vegetable
phenylacetonitrile	0	0	0	316,086	29.51	nr
5*H*-furan-2-one	0	0	0	286,753	14.63	buttery
3-methylfuran	0	0	0	258,698	2.97	nr
2-(furan-2-ylmethyl)furan	0	0	0	220,455	13.87	roasted
5-(formylfuran-2-yl)methyl acetate	0	0	0	192,403	19.99	nr
2-(furan-2-ylmethyl)-5-methylfuran	0	0	0	168,726	16.32	nr
2-oxopropyl acetate	0	0	0	130,636	7.08	fruity, buttery, dairy
ethyl 3-methylbutanoate	0	0	0	123,878	28.04	fruity
5-methyl-2(5*H*)-furanone	0	0	0	99,098	13.48	nr
6-(5-methyl-furan-2-yl)-hexan-2-one	0	0	0	73,787	18.38	nr
2-(4-aminophenyl) acetonitrile	0	0	0	72,461	17.02	nr
4-morpholin-4-yl aniline	0	0	0	68,907	30.07	nr
2-propylpyrazine	0	0	0	68,501	11.37	brothy, sulfury, smoky, beany
5-(furan-2-ylmethyl)-5-methylfuran-2-one	0	0	0	68,166	22.16	nr
methylpyridine-3-carboxylate	0	0	0	67,852	15.69	herbal, tobacco
1-(2-hydroxy-5-methylphenyl)ethanone	0	0	0	60,350	17.94	floral
2-methyl-furan-3-carboxylic acid n′-acetyl-hydrazide	0	0	0	55,314	22.98	nr
4-methyl-2*H*-quinolin-5-one	0	0	0	47,832	26.54	nr
1*H*-pyrrolo [2,3-*b*]pyridine	0	0	0	43,278	17.04	nr
2,5-dimethylfuran	0	0	0	42,981	3.82	meaty
1-thiophen-3-ylethanone	0	0	0	39,659	13.48	nr
thiophene-3-carbaldehyde	0	0	0	34,872	10.65	nr
*trans*-isoeugenol	0	0	0	33,906	23.72	spicy
2-acetylcyclohexan-1-one	0	0	0	30,995	16.09	nr
1-(5-methylthiophen-2-yl)ethanone	0	0	0	30,609	17.06	floral
nonanoic acid	0	0	0	30,342	19.64	waxy
1-(4-hydroxyphenyl)propan-1-one	0	0	0	27,581	18.74	nr
2-methyl-5,6,7,8-tetrahydroquinoxaline	0	0	0	26,835	19.60	animal
6-methyl-2*H*-1-benzopyran-2-one	0	0	0	26,168	26.49	coconut
5,6,7,8-tetrahydroquinoxaline	0	0	0	25,822	17.16	nr
4-nitrophenyl pentanoate	0	0	0	24,919	17.48	nr
6-nonyloxan-2-one	0	0	0	23,085	34.22	waxy
2,5-dimethyl-3-(2-methylprpyl) pyrazine	0	0	0	22,126	17.23	nr
1-thiophen-2-ylethanone	0	0	0	20,122	13.25	onion
2-methyl-5-[(5-methylfuran-2-yl)methyl]furan	0	0	0	19,517	19.09	nr
1,3-thiazole	0	0	0	19,465	4.49	green, nutty, tomato
*cis*-dehydroxy linalool oxide	0	0	0	18,958	10.85	floral green
(3*E*)-3,7-dimethylocta-1,3,6-triene	0	0	0	18,666	12.09	nr
2,3,5-trimethyl-6-prop-2-enyl pyrazine	0	0	0	18,344	20.39	nr
7-methyl-3-methylene-octa-1,6-diene	0	0	0	18,224	10.34	spicy, wood
3-ethylpyridine	0	0	0	17,528	18.05	tobacco
3-methylpyridine	0	0	0	15,821	7.32	green
1-pyridin-4-ylethanone	0	0	0	15,585	13.60	burnt
1-(5-methylfuran-2-yl)butan-1-one	0	0	0	15,300	17.41	nr
2-[(4-ethylphenoxy)methyl]oxirane	0	0	0	15,066	20.55	nr
1-(4-methylthiophen-3-yl)ethanone	0	0	0	13,022	15.35	nr
1-ethyl-4-methoxy-9*H*-pyrido [3,4-*b*]indole	0	0	0	12,922	31.55	nr
2,6,6-trimethyl-2-cyclohexene-1,4-dione	0	0	0	12,523	15.01	musty, citrus
1-(4*H*-pyridin-1-yl)ethanone	0	0	0	12,011	11.22	nr
1-(4-methylthiophen-2-yl)ethanone	0	0	0	11,558	15.29	nutty
2-methylcyclopent-2-en-1-one	0	0	0	10,775	8.01	nr
naphthalene	0	0	0	9395	17.02	mothballs
propanoic acid	0	0	0	8969	4.32	nr
1-methyl-4-(propan-2-y)lbenzene	0	0	0	7475	11.88	spicy, citrus
2,4-dimethyl-1,3-thiazole	0	0	0	6090	7.46	briney, sulfury, burnt, rubber, medicine

### 2.7. Compounds that Persisted through Roasting

There were 72 common compounds among the four coffee samples (Figure 4. Table 3) that survived roasting linking the green and roasted coffees grown in this region. Of the 72, 48 compounds exhibited a 2-fold or higher increase in the concentration ratio between the Green and Roasted AAA coffees, and 65 were 2-fold or greater ratio in the Green AA to Roasted AAA coffee concentration ratio. Among them (48 and 65), 40 compounds were common to both quality levels. Toluene is the only compound in the green coffees that decreased and was measurable in the two roasted coffees. Many of the 72 common compounds have a sensory impact in green coffee [39], including linalool (*floral*), nonanal (*rose, fruit*), and phenylacetaldehyde (*floral, honey*). Additionally, linalool has been shown to be specific marker for coffee processing [40], and, thus, is likely to be highly related to overall coffee quality.

**Table 3 molecules-27-05328-t003:** Compounds that survived roasting.

Compound	Green AAA	Green AA	Roast AAA	Roast AA	Retention Time	Lit Odor
hexadecanoic acid	1,108,940	6200	233,280	59,944	37.27	waxy
toluene	830,552	107,867	183,937	76,509	4.83	nr
pyridine	501,494	13,372	1,495,635	452,851	4.48	resinous, roasted, burnt
benzophenone	376,506	20,060	296,941	40,875	29.49	balsamic
1-(furan-2-yl)ethan-1-one	269,574	103,844	684,350	128,217	8.42	fruity sweet, caramel
nonanal	231,092	5426	314,988	26,032	14.54	waxy, rose, orange
decanal	144,208	36,319	193,021	20,118	17.70	aldehydic
734	131,831	31,844	181,302	162,176	22.10	
2-methyl-1-(1,1-dimethylethyl)-2-methyl-1,3-propanediyl ester	127,289	4785	210,279	39,256	28.59	nr
2-ethenyl-6-methylpyrazine	111,984	4900	76,380	105,157	11.64	roasted, potato
furan-2-carbaldehyde	97,096	5747	1,026,548	67,410	6.36	woody, almond, baked bread
5-methylfuran-2-carbaldehyde	93,410	7790	2,726,476	35,266	10.02	caramel-like, bready, coffee-like
378	90,465	51,972	830,442	89,822	22.12	
2-ethenyl-5-methylpyrazine	72,870	5200	51,693	68,303	11.71	rubber, smoky, chemical, greasy, onion
449	69,464	16,976	126,788	53,076	26.66	
2,4-ditert-butylphenol	67,202	24,294	53,059	39,125	26.44	nr
furan-2-ylmethyl acetate	64,927	17,522	2,093,291	5186	11.00	fruity sweet, banana-like
octanal	64,004	64,330	69,829	38,039	11.21	waxy, citrus, green, fatty
2-ethylpyrazine	60,417	761,061	535,929	761,061	8.50	nutty
2-methylpyrazine	47,786	8794	734,866	246,899	6.08	nutty
(*E*,*E*)-2,4-heptadien-6-ynal	44,850	44,963	108,827	114,800	9.89	nr
2-ethyl-1-hexnol	39,167	45,579	39,044	88,651	12.07	citrus, floral
420	34,364	6842	89,710	4916	26.82	
506	33,761	4750	218,646	193,545	25.37	
4-ethenyl-1,2-dimethoxybenzene	33,129	4965	648,094	596,689	22.49	floral, green
phenol	31,362	4808	67,790	571,972	10.74	medicinal, tar, phenolic
linalool	30,870	11,402	78,965	98,374	14.40	fruity, floral
1,3-benzothiazole	30,124	41,072	33,267	72,051	18.31	meaty
1-phenylethan-1-one	30,092	130,549	48,599	1,207,900	13.31	floral
2,2′-(oxydimethylene)difuran	26,876	58,142	597,198	131,155	20.61	coffee, nutty, earthy
331	26,456	10,222	319,210	151,125	19.41	
514	26,220	11,153	561,508	166,993	27.00	
1,3-xylene	24,927	14,683	33,280	16,454	7.19	nr
1*H*-indole	24,822	7635	258,353	30,262	20.41	fecal
4-ethyl-2-methoxyphenol	18,664	4659	475,344	131,434	19.94	balsamic, clove, phenolic, woody, smoke,
2-(2-propenyl)furan	18,134	18,360	8514	18,360	6.96	nr
2-phenylacetaldehyde	17,357	15,093	82,885	530,637	12.56	green, floral, honey, cocoa
619	16,717	9832	180,995	104,992	31.06	
styrene	16,637	14,947	22,648	14,979	7.78	nr
751	16,394	4650	89,238	56,635	41.24	
1-(1-methyl-1*H*-pyrrol-2-yl)-ethanone	15,794	4263	253,898	62,326	13.58	musty
p-methylacetophenone	14,685	67,536	139,498	802,460	17.08	na
471	14,261	5482	442,113	5482	24.69	
489	13,635	6069	142,306	34,194	26.40	
382	12,689	24,900	90,967	223,435	23.03	
277	11,900	8253	498,094	500,537	17.10	
1-(furan-2-yl)propan-1-one	11,781	9982	281,461	296,132	11.42	fruity
681	11,748	14,780	464,226	646,418	33.22	
3-pheny pyridine	11,322	33,697	72,022	70,303	25.13	nr
heptan-2-one	11,163	24,213	21,113	44,952	7.81	cheesy
1-methyl-4-(prop-1-en-2-yl)cyclohex-1-ene	10,476	9440	19,822	21,809	12.01	citrus
1,2-xylene	10,334	20,449	17,638	31,242	7.83	nr
5-methyl-6,7-dihydro-5*H*-cyclopenta[*b*]pyrazine	9832	12,867	86,032	127,990	15.71	nr
3,4-dimethyl-1*H*-pyrrole-2-carboxaldehyde	9744	17,959	163,351	1,435,806	16.27	nr
132	9723	6481	15,159	18,572	13.42	
methyl salicylate	9104	9255	75,020	98,271	17.39	wintergreen
198	8682	13,196	46,484	48,733	13.71	
furan-2-ylmethyl propanoate	8673	4994	296,194	270,827	14.00	nr
490	8517	9901	336,296	462,642	27.02	
(6*E*,8*E*)-megastigma-4,6,8-trien-3-one	8268	13,727	71,780	479,280	29.44	nr
dodecanal	7960	10,237	18,759	167,465	23.61	aldehydic
1-(4-hydroxyphenyl)-2-methylpropan-1-one	7565	5897	130,450	108,157	22.50	nr
3,7-dimethyl-6,7-dihydro-5*H*-cyclopentapyrazine	7277	9821	49,083	91,792	18.24	nr
2-[(methyldithio) methyl]-furan	7261	15,630	123,674	128,211	17.92	sulfury
1-(pyridin-2-yl)ethanone	7225	6667	24,502	20,747	12.23	popcorn
1-(2-hydroxyphenyl)ethanone	7223	36,783	65,120	63,927	16.40	phenolic
2-methoxyphenol	7107	17,135	124,736	204,167	14.08	phenolic, woody
6-methoxy-2-methylquinoline	7060	6876	39,934	76,719	24.52	nr
15	6361	12,780	7646	12,870	3.32	
2-phenylbut-2-enal	4983	58,614	135,175	61,768	19.78	musty, green
2-methylpyridine	4886	4923	8723	9524	6.04	sweat
733	4012	8892	132,975	479,806	18.86	

### 2.8. Relevance of GC-GC/MS for Natural Product Profiling

GC-GC/MS with spectral deconvolution and MS subtraction is an especially powerful technique for speciating organics in complex natural products. By overloading the first column and relying on the second column to separate 1-min sample fractions, nearly 1000 compounds were detected. Given that GC/MS employing high-resolution mass spectrometers is limited by mass injected on-column, an equal level of speciation would not be possible [10]. In this study, approximately 1000 versus 500 compounds were uniquely detected by GC-GC/MS compared to GC/MS.

Another advantage of GC-GC/MS is the ability to heartcut compounds of importance. For example, the same instrument can be used to identify aroma compounds, both pleasant and foul, via the olfactory port, and then trap an individual compound by heartcut onto an adsorbent attached to the sniffing port. Sufficient mass of individual components can be collected for additional analysis by NMR, UV, IR, etc., which can support tentative identification of unknowns. Work is in progress to illustrate this feature. The major limitation in GC-GC/MS acceptance as a routine analytical tool is the amount of time it takes to completely profile a product and to build a target a library, which typically requires 3.5 days of instrument runtime and a few days of library-building. The aim of the Ion Analytics workflow is to automate the data processing aspect of library-building, and with constructed target libraries employ GC/MS with either low- or high-resolution spectrometers to profile new samples.

Our approach allowed us to measure the differences in concentration between and among different quality green and roasted coffees, which should provide the ability to authenticate coffee by quality and/or region as we did for tea [11]. For example, coffee quality can be evaluated at various points along the supply chain, and higher quality designations will bring higher prices. Measuring cherry ripeness at harvest is one of the initial metrics of quality commonly used by produces around the world. Often, superior quality cherry with higher and consistent ripeness will be processed differently or simply tracked and kept separate, with a premium charged when the green coffee is sold for export. Further work will reveal how and by what means detailed chemical characteristics of coffee can be used to differentiate quality. In this study, we evaluated quality designations applied to the ripe coffee cherry by the Coopedota Cooperative in Tarrazú, Costa Rica, and found more than 100 chemical markers that differentiate the two highest quality (AAA and AA) coffees.

## 3. Materials and Methods

### 3.1. Workflow

#### 3.1.1. Database Building

This study follows a similar GC-GC/MS library-building process, as previously described [3,11]. The 1-min sample portions (heartcuts) were transferred from the first to the second column (Figure 1), with each subsequent injection made only after the preceding sample fraction eluted from both columns. A total of 44 data files were obtained. The GC-GC/MS analysis time lasted 3 days and, although long, resulted in chromatography that yielded clean mass spectra.

IA was used to create the database. In heartcut 1, the software inspected each peak to determine if the corresponding peak scans were constant. If so, coffee organics were tentatively identified by comparing MS fragmentation patterns and retention indices (RI) to those found in mass spectral libraries, such as NIST17 and Adams’ Essential Oil Library. Positive compound confirmation was made by comparing the tentatively identified compound spectra and RI’s to more than 400 reference standards. If neither positive nor tentative identification was possible, a numerical identifier was assigned so that the compound’s relative concentration can be compared across samples. Additionally, we added each compound’s CAS # (when applicable), retention time (index), mass spectrum (used in MS subtraction), and 3–6 target ions and relative abundances (used in spectral deconvolution) to the database.

If peak scans were not constant, the IA software selected 3–5 invariant scans, averaged them, and then subtracted that spectrum from the total ion current signal. If the resulting ion signals were constant, tentative identification was made as described above. If, after MS subtraction, the resulting ion signals approximated background noise, no additional action was needed. If not, the workflow continued until the resulting ion signal approximated background signals. When all of the Roasted AAA heartcuts were analyzed, the initial database contained the identity of 750 compounds, their retention, and spectral data [11].

#### 3.1.2. Target Compound Identification

Ion Analytics was used to extract at least three ions, viz., the main (100%) ion and at least two qualifier ions, for each target compound. Confirming ions were normalized to the main ion according to Equation (1), with the expected deviation ≤ 20 % for at least five consecutive scans.
(1)$$I_{i}\left(t\right)=\frac{A_{i}\left(t\right)}{R_{i}A_{i}}$$
$I_{i}\left(t\right)$ is the reduced ion intensity relative to the main ion, i=1 at scan (t);$A_{i}\left(t\right)$ is the absolute *i-th* confirming ion intensity at scan (t);$R_{i}$ is the expected relative ion abundance ratio for the *i-th* ion;$A_{i}$ is the absolute abundance of the main ion.

A histogram representing the normalized ion ratios was generated for each scan; the flatter it is, the closer the actual ion ratio is to the expected ratio for that scan. The spectral match is determined by calculating the average reduced intensity deviation (ΔI) of each of the N confirming ions (Equation (2)). The closer (ΔI) is to zero, the better the match.
(2)$$\Delta I=\frac{\sum_{i=1}^{N-1}\sum_{j=i+1}^{N}Abs\left(I_{i}-I_{j}\right)}{\sum_{i=1}^{N-1}i}$$

Criteria 1 and 2 are met when ΔI ≤ K + Δ_0_/A_i_, where K is the user defined (acceptable) relative percent difference and Δ_0_ is the additive error attributable to instrument noise and/or background signal. The scan-to-scan variance (SSV) is calculated from Δ*E* = Δ*I**log (*A*_i_). Δ*E* is the scan-to-scan variance (SSV). It is acceptable when Δ*I* or Δ*E* are below the maximum allowable error, Δ*E*, max, which was set at 7 for this study. The algorithm calculates the relative error by comparing the mass spectrum of one scan against the others. For example, the first is compared to scans 2, 3, 4…n. The second to the 3rd, 4th, 5th …n and (n − 1) to the nth scan. The smaller the difference, the closer the SSV is to zero, the better the spectral agreement.

Criteria 3 is the Q-value, which measures the total ion ratio deviation of the absolute value of the expected minus observed ion ratios divided by the expected ion ratio times 100 for each ion across the peak. The closer the value is to 100, the higher the certainty between sample and reference, library, and/or literature spectra. The Q-ratio is the final criteria. It compares the molecular and confirming ion intensity ratios across the peak. In this study the Q-value and Q-ratio acceptability limits were ≥95% and ±20%, respectively. These four criteria formed a single criterion and was used to affirm compound identity. When the criterion is met, the software normalizes confirming ions to the main ion producing a histogram of ion signals. Visual inspection of target compound histograms makes compound identity easy to affirm. Spectral deconvolution with MS subtraction provided the means to identify detectable compounds by GC/MS.

### 3.2. Coffee Roasting

For this study, the AAA and AA coffees were obtained from the cooperative, and then half of each batch was roasted by the cooperative using a Probat BRZ 2 sample roaster. Once roasted, the four samples (AAA Green, AA Green, AAA Roasted, and AA Roasted) were shipped to Tufts University and stored at −20 °C until analyzed. Note, for this study we reference the raw, dried coffee as “green” coffee, and roasted coffee as such.

### 3.3. Coffee Extraction

For GC/MS analyses, each of the four coffee samples were subdivided into three portions. From each portion a 23.5 g sample was coarsely ground using a commercial grinder (setting 8, Mahlkönig model # KS32/100, Durham, NC, USA), and then brewed with 355 mL of water under slightly reduced pressure, 90 kPa at 91 °C, using the BKON (Mooretown, NJ, USA) Rain technology extraction system (https://www.bkonbrew.com/rain, accessed on 18 July 2022). Once brewed, 1 mL was diluted to 10 mL using Aquafina water in 10 mL vials. Organics were sorbed onto Twister^®^ stir-bars (coated with PDMS) for 1 h at 1200 rpm. After extraction, stir-bars were removed from the vials and 1 uL of internal standard (10 mg/L d8–napthlene) was added directly onto each stir-bar. The stir-bars were placed into glass desorption tubes and analyzed using the selectable 1D/2D GC/MS.

### 3.4. Chemical Standards

The retention index (RI) of each compound was calculated using a standard mixture of C7 to C30 n–alkanes obtained from Sigma-Aldrich (St. Louis, MO, USA). More than 400 reference standards were obtained from Sigma-Aldrich, TCI America (Portland, OR, USA), Acros Organics (Pittsburgh, PA, USA), Alfa Aesar (Ward Hill, MA, USA), MP Biomedicals (Santa Ana, CA, USA), SPEX CertiPrep (Metuchen, NJ, USA), and AccuStandard (New Haven, CT, USA). Reference standard retention times were analyzed based on their elution times on the 2nd column (RXI-5MS, Restek, Bellefonte, PA, USA).

### 3.5. Selectable 1D/2D GC/MS

Two low thermal mass (LTM) column modules (Agilent Technology, Santa Clara, CA, USA), connected by a Deans switch inside the GC oven, heated the columns. Flow through the Deans switch and 3-way splitter provided the means to operate the instrument as a selectable 1D/2D multi-detection system. When samples flowed through the splitter to the MS and flame ionization detector, the former was used to identify/quantify analytes, the latter to monitor heartcut consistency (Figure 5; black arrow). When samples flowed through the splitter to the MS and olfactometer detection port (ODP3, Gerstel, Mulheim, Germany), analytes were identified/quantified and, when warranted, sniffed for sensory attributes (Figure 5; red arrow). When samples flowed through the Deans switch directly to the splitter, the instrument operated as a GC/MS or GC/MS-ODP (Figure 5; green arrow).

An Agilent model 7890A/5975C GC/MS (Agilent Technology, Santa Clara, CA, USA) housed the LTMs, flow switches, and transfer lines. Agilent’s pneumatics controller module (PCM) and software controlled the flow between columns to make heartcuts. The instrument was equipped with a multi-purpose sampler (MPS, Gerstel, Linthicum, MD, USA), thermal desorption unit (TDU, Gerstel, Linthicum, MD, USA), cooled injection system (CIS4, Gerstel, Linthicum, MD, USA), and cryotrap/thermal desorption unit (CTS2, Gerstel, Linthicum, MD, USA). The MPS automated the stir bar injection process while the CTS2 freeze-trapped, and then desorbed each sample fraction transferred from the first to the second column. LTM 1 housed column 1 (C1, 30 m × 250 µm × 0.25 µm RTX-Wax, Restek, Bellefonte, PA, USA). Operating conditions were: initial temperature 40 °C (1 min), temperature program 240 °C at 5 °C/min. LTM 2 housed column 2 (C2, 30 m × 250 µm × 0.25 µm RXI-5MS, Restek). C2 operating conditions were: initial 40 °C (1 min), ramping to 300 °C at 5 °C/min. MS ion source and quadrupole temperatures were 230 °C and 150 °C, respectively. The electron impact voltage and *m*/*z* scan range and frequency were 70 eV and 50–350 and 12 scans/s. Spectra were collected in positive ion mode. GC/MS measurements are based on relative abundances of analyte to internal standard, with positive detection based on ratios > 5000 units, see Table 1.

## Figures and Tables

**Figure 1 molecules-27-05328-f001:**
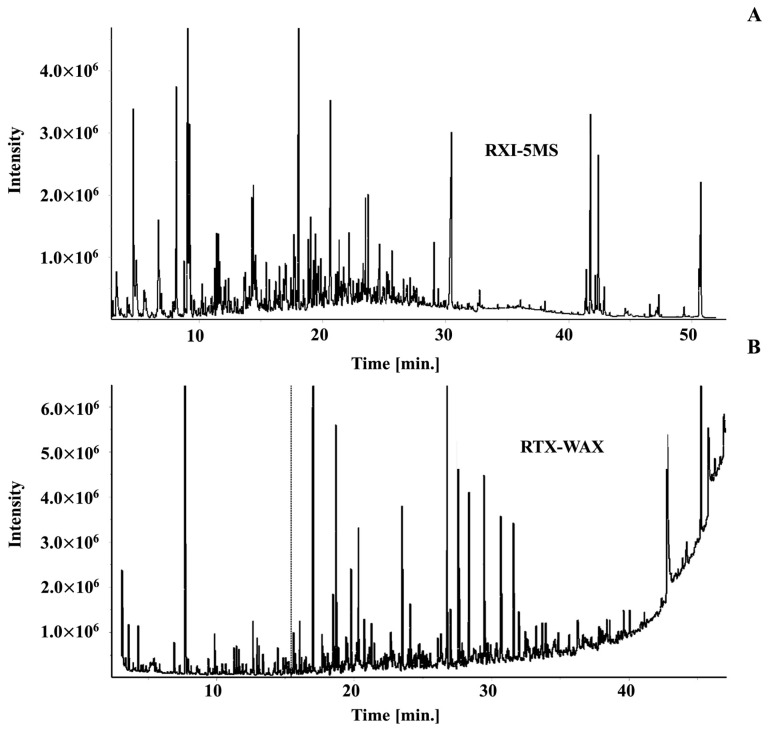
GC/MS chromatogram of roasted coffee on (**A**) non-polar (RXI-5MS) and (**B**) polar (RTX-WAX) phases.

**Figure 2 molecules-27-05328-f002:**
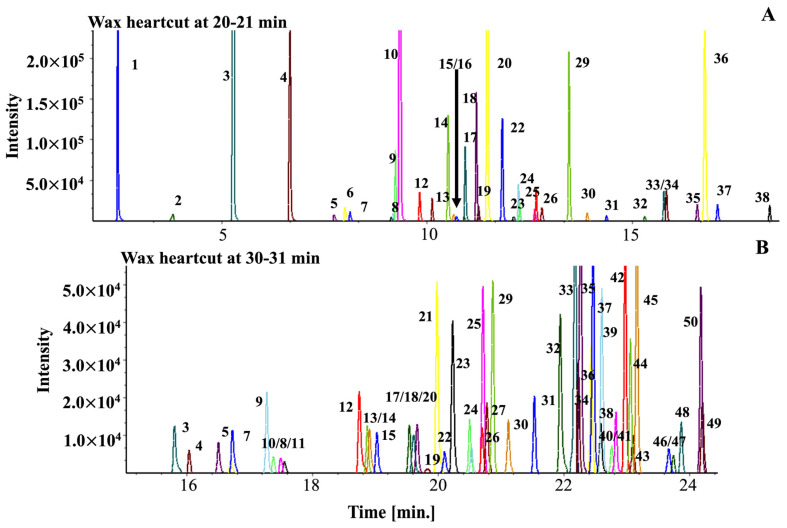
Sample GC-GC/MS total ion chromatograms of 1-min sample fractions transferred from the polar phase column (RTX-WAX) to the non-polar column (RTX-5MS). (**A**) Total ion current chromatogram collected from a heart cut at minute 20–21. (**B**) Total ion current chromatogram collected from a heart cut at minute 30–31.

**Figure 3 molecules-27-05328-f003:**
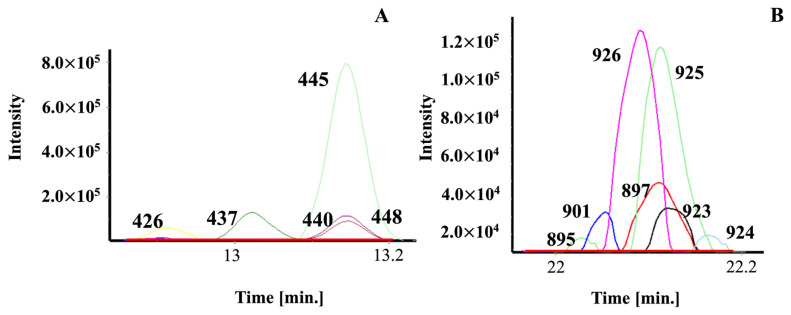
Illustrative examples of spectral deconvolution and MS subtraction using the GC-GC/MS database as target compounds. Each panel displays a portion of a 1D GC (RTX-5MS column)/MS analysis. (**A**) peak ID’s and heartcuts time (RTX-WAX): 426 = 1-methylethenyl-pyrazine, 20–21 min; 437 = 1-(1-methyl-1*H*-pyrrol-2-yl)-ethanone, 29–30 min; 440 = m-cresol, 30–31 min; 445 = 2,6-diethyl-pyrizine, 16–17 min; and 448 = 3-ethyl-2,5-dimethyl-pyrizine, 19–20 min. (**B**) Peak ID’s and heartcuts: 895 = #354, 31–32 min; 897 = 2-(hydroxymethyl)-benzoic acid, 29–30 min; 901 = 4-ethenyl-1,2-dimethoxy-benzene; 29–30 min; 923 = 5-hydroxy-3,3-dimethyl-1-benzofuran-2-one, 40–41 min; 924 = 4-methylindole, 37–38 min; 925 = 1-(4-hydroxyphenyl)-2-methyl-1-propanone, 37–38 min; and 926 = #381, 37–38 min.

**Figure 4 molecules-27-05328-f004:**
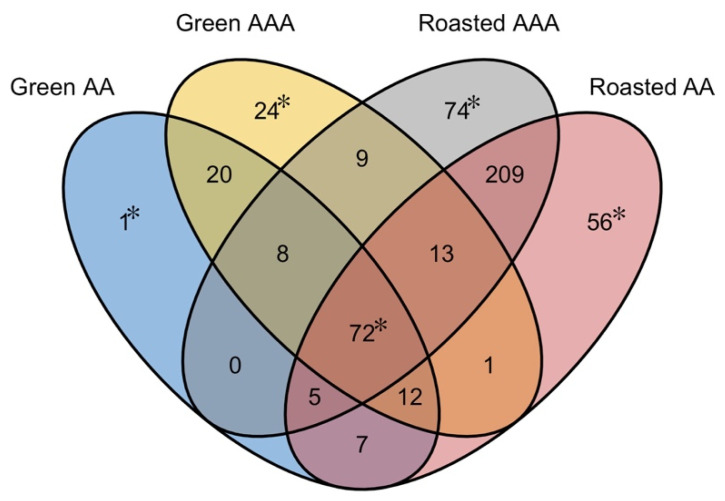
Venn diagram showing the commonality of identified compounds. Shared compounds are indicated in the overlapping areas of each oval; for example, 112 compounds were found common between the Green AAA and Green AA, which is determined by adding 20, 8, 72, and 12. Sets labeled with “*” are listed in Table 1, Table 2 and Table 3.

**Figure 5 molecules-27-05328-f005:**
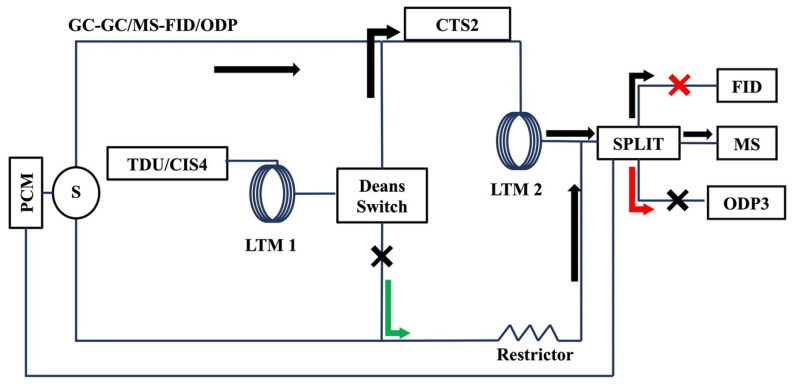
Selectable 1D/2D GC/MS-FID/ODP. Black arrows indicate sample separation by GC-GC for library-building purposes. Red arrows indicate flow the path when operating the ODP. The 1D flow path does not include the LTM2 and is indicated with the green arrow.

## Data Availability

Contact S.F. regarding data availability.

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
