# Peer review of "The Chemistry of Green and Roasted Coffee by Selectable 1D/2D Gas Chromatography Mass Spectrometry with Spectral Deconvolution"

_molecules, 2022, doi:10.3390/molecules27165328_

Round 1

Reviewer 1 Report

The current manuscript by Frost and coworkers describes the chemical differences between two grades of green coffee, pre and post roasting, via GCMS and GC-GC/MS analysis. Specifically, authors utilize Ion analytics software to deconvolute complex mixtures of coffee bean extracts, finding 72 compounds that survive the roasting process that can be used to discriminate between the two studied grades of coffee. The experimental design and results are well documented, however I have a few concerns:

1. The abstract claims that the 72 compounds that survive the roasting process can “be used to discriminate roasting methods.” However, I do not see multiple roasting methods described nor do I see any mention of testing two or more roasting methods. I see data to support that those compounds could be used to discriminate the grade of coffee pre-roasting. Please address.

2. These instruments are incredibly specialized and are not standard instruments found in university core facilities, academic labs, or industrial laboratories. To enhance the impact of the manuscript, I would suggest adding a discussion related to the benefits and draw-backs of purchasing/using a GC-GC/MS system vs a high-resolution GCMS, such a GC-orbitrap. Several reviews exist comparing low- and high-resolution GCMS data, so it follows that this could be used to address how a GC-GC/MS system would likely perform relative to a high-resolution system under similar circumstances.

3. Based on my familiarity with GC-GCMS systems, it is more common that the second column is considerably shorter than the first, in the range of a single to a few meters. This method considers two equal length columns which will surely enhance the separation of the mixture. It would be prudent to address this beyond the methods, especially considering the three day acquisition time. One could also argue that a GCMS with autosampler and ~40 minute method with 1 minute fractions, followed by second injection of each fraction over a secondary column could be conducted in a day less time overall. This is not to say that software does not integrate those data better using the system in the study, but it does warrant is discussion.

Author Response

Response to Review: Molecules-1848749 The Chemistry of Green and Roasted Coffee by Selectable 1D/@d Gas Chromatography Mass Spectrometry with Spectral Deconvolution.

We thank the reviewer for their thoughtful comments and suggestions, which we think will improve the clarity and impact of our work. We have addressed the following:

Review 1

The current manuscript by Frost and coworkers describes the chemical differences between two grades of green coffee, pre and post roasting, via GCMS and GC-GC/MS analysis. Specifically, authors utilize Ion analytics software to deconvolute complex mixtures of coffee bean extracts, finding 72 compounds that survive the roasting process that can be used to discriminate between the two studied grades of coffee. The experimental design and results are well documented; however I have a few concerns:

  1. The abstract claims that the 72 compounds that survive the roasting process can “be used to discriminate roasting methods.” However, I do not see multiple roasting methods described nor do I see any mention of testing two or more roasting methods. I see data to support that those compounds could be used to discriminate the grade of coffee pre-roasting. Please address.

The reviewer is correct. It appears we did not make ourselves as clear as we should have. The compounds that survive roasting do indeed differentiate green coffee quality but have the potential to be indicators of coffee quality after roasting. Work is in progress to evaluate whether the same set of indicator compounds can be used to assign coffee quality after roasting the same beans under different conditions. We have clarified the manuscript at line 19 – 20 as follows:

“Of these, 72 compounds survive the roasting process and can be used to discriminate green coffee quality after roasting.”

  1. These instruments are incredibly specialized and are not standard instruments found in university core facilities, academic labs, or industrial laboratories. To enhance the impact of the manuscript, I would suggest adding a discussion related to the benefits and draw-backs of purchasing/using a GC-GC/MS system vs a high-resolution GCMS, such a GC-orbitrap. Several reviews exist comparing low- and high-resolution GCMS data, so it follows that this could be used to address how a GC-GC/MS system would likely perform relative to a high-resolution system under similar circumstances.

We agree with the reviewer and have added a new section, please see lines 265 to 306 of the revised manuscript:

2.8 Relevance of GC-GC/MS for natural product profiling

            GC-GC/MS with spectral deconvolution and MS subtraction is an especially powerful technique for speciating organics in complex natural products. By overloading the first column and relying on the second column to separate 1-min sample fractions nearly 1000 compounds were detected. Given that GC/MS employing high resolution mass spectrometers is limited by mass injected on-column, and thus an equal level of speciation would not be possible. [10]. In this study, approximately 1000 versus 500 compounds were uniquely detected by GC-GC/MS compared to GC/MS. The ability to discern low concentration analytes in one sample and add them to the database is a critically important advantage as we found in our tea study. Low concentration metabolites found in pre-monsoon teas increased in concentration during the monsoon rains. Profiling metabolites as a function of changing climate conditions is important to both farmers and consumers alike [3, 12, 13].

Another advantage of GC-GC/MS is the ability to heartcut compounds of importance. For example, the same instrument can be used to identify aroma compounds, both pleasant and foul, via the olfactory port, and then trap an individual compound by heartcut onto an adsorbent attached to the sniffing port. Sufficient mass of individual components can be collected for additional analysis by NMR, UV, IR, etc. which will support tentative identification of unknowns by mass spectrometry. Work is in progress to illustrate this feature. The major limitation in GC-GC/MS acceptance as a routine analytical tool appears to be the amount of time it takes to completely profile a product and to build a target a library, which typically requires 3.5 days of instrument runtime and a few days of library-building. The aim of the Ion Analytics workflow is to automate the data processing aspect of library-building and once constructed employ GC/MS with either low- or high-resolution spectrometers to profile new samples.

Our approach allowed us to measure the differences in concentration between and among different quality green and roasted coffees, which should provide the ability to authenticate coffee by quality and/or region as we did for tea[11]. For example, coffee quality can be evaluated at various points along the supply chain, and higher quality designations will bring higher prices. Measuring cherry ripeness at harvest is one of the initial metrics of quality commonly used by produces around the world. Often, superior quality cherry with higher and consistent ripeness will be processed differently or simply tracked and kept separate, with a premium charged when the green coffee is sold for export. Further work will reveal how and by what means detailed chemical characteristics of coffee can be used to differentiate quality. In this study, we evaluated quality designations applied to the ripe coffee cherry by the Coopedota Cooperative in Tarrazú, Costa Rica, and found more than 100 chemical markers that differentiate the two highest quality (AAA and AA) coffees.

  1. Based on my familiarity with GC-GCMS systems, it is more common that the second column is considerably shorter than the first, in the range of a single to a few meters. This method considers two equal length columns which will surely enhance the separation of the mixture. It would be prudent to address this beyond the methods, especially considering the three day acquisition time. One could also argue that a GCMS with autosampler and ~40 minute method with 1 minute fractions, followed by second injection of each fraction over a secondary column could be conducted in a day less time overall. This is not to say that software does not integrate those data better using the system in the study, but it does warrant is discussion.

            2D GC systems based on 1 long and 1 short, narrow bore column is known as comprehensive GCxGC/MS. We have published several papers comparing/contrasting GCxGC/MS with GC-GC/MS with and without the Ion Analytics software. First, mass on-column is limited in GCxGC/MS for the same reason GC/MS is limited. Second, although separation efficiency is higher for GCxGC (multiplicative) compared to GC-GC (additive), olfactory analysis and analyte collection is impossible due to the fact that the first column peak is cut into many peaks on the second column. The band width of these peaks is so narrow that it is not possible to clear your nose from one peak-to-the-next to correctly assign aroma.

The reviewer’s arithmetic is unclear. The system we described employs an autosampler that makes 40 injections to span the 40 min separation time on the first column based on 1-min sample fractions. If the second column is also a 40-min separation, one still needs to complete that separation before making the next injection or highly volatile organics from a subsequent injection will catch up to a low volatility organic from the preceding injection; this is called wraparound. It is why GCxGC approaches the limit for heartcutting and offers higher separation efficiency than GC-GC. To shorten runtimes in GC-GC, one can temperature ramp the first column after heartcutting but this approach is somewhat offset by the time it takes to readjust flow rates. These issues are well-known in the field of separation science, AR has discussed it in other publications; further discussion here would distract from the aim of this paper.

Reviewer 2 Report

The authors present a work entitled The Chemistry of Green and Roasted Coffee by Selectable 1D / 2D Gas Chromatography Mass Spectrometry with Spectral Deconvolution ". The manuscript is well written and well organized. The experiments are clearly reported and the references are congruous. The topic it certainly concerns a relevant research topic and can be greatly appreciated by readers. The possibility of using software and databases for the characterization and identification of the analytes today represents an important and much debated topic in the scientific world. Likewise, the application of specific spectral deconvolution plays a central role in the search for new analytes in the various matrices.

For these reasons, I believe the work can be accepted for publication on Molecules. In order to improve the quality of the work, I only suggest the following variations / corrections:

·     According to the above, Roasted AAA should be reported in the sentence "This is indicated by the number 8 in the overlapping portion of the three ovals for Green AA, Green AAA, and Roasted AA"

·      It would be useful to insert a small paragraph with the main conclusions, also indicating what the analytical advantages of the proposed method could be for future works

Author Response

Response to Review: Molecules-1848749 The Chemistry of Green and Roasted Coffee by Selectable 1D/@d Gas Chromatography Mass Spectrometry with Spectral Deconvolution.

We thank the reviewer for their thoughtful comments and suggestions, which we think will improve the clarity and impact of our work. We have addressed the following:

Review 2

The authors present a work entitled The Chemistry of Green and Roasted Coffee by Selectable 1D / 2D Gas Chromatography Mass Spectrometry with Spectral Deconvolution ". The manuscript is well written and well organized. The experiments are clearly reported and the references are congruous. The topic it certainly concerns a relevant research topic and can be greatly appreciated by readers. The possibility of using software and databases for the characterization and identification of the analytes today represents an important and much debated topic in the scientific world. Likewise, the application of specific spectral deconvolution plays a central role in the search for new analytes in the various matrices.

For these reasons, I believe the work can be accepted for publication on Molecules. In order to improve the quality of the work, I only suggest the following variations / corrections:

  1. According to the above, Roasted AAA should be reported in the sentence "This is indicated by the number 8 in the overlapping portion of the three ovals for Green AA, Green AAA, and Roasted AA"

Thank you, we have modified the manuscript at line 188 - 189 as follows:

“This is indicated by the number 8 in the overlapping portion of the three ovals for Green AA, Green AAA, and Roasted AAA.”

  1. It would be useful to insert a small paragraph with the main conclusions, also indicating what the analytical advantages of the proposed method could be for future works

We agree with the reviewer and have added a new section, please see lines 265 to 306 of the revised manuscript:

        2.8 Relevance of GC-GC/MS for natural product profiling

            GC-GC/MS with spectral deconvolution and MS subtraction is an especially powerful technique for speciating organics in complex natural products. By overloading the first column and relying on the second column to separate 1-min sample fractions nearly 1000 compounds were detected. Given that GC/MS employing high resolution mass spectrometers is limited by mass injected on-column, and thus an equal level of speciation would not be possible. [10]. In this study, approximately 1000 versus 500 compounds were uniquely detected by GC-GC/MS compared to GC/MS. The ability to discern low concentration analytes in one sample and add them to the database is a critically important advantage as we found in our tea study. Low concentration metabolites found in pre-monsoon teas increased in concentration during the monsoon rains. Profiling metabolites as a function of changing climate conditions is important to both farmers and consumers alike [3, 12, 13].

Another advantage of GC-GC/MS is the ability to heartcut compounds of importance. For example, the same instrument can be used to identify aroma compounds, both pleasant and foul, via the olfactory port, and then trap an individual compound by heartcut onto an adsorbent attached to the sniffing port. Sufficient mass of individual components can be collected for additional analysis by NMR, UV, IR, etc. which will support tentative identification of unknowns by mass spectrometry. Work is in progress to illustrate this feature. The major limitation in GC-GC/MS acceptance as a routine analytical tool appears to be the amount of time it takes to completely profile a product and to build a target a library, which typically requires 3.5 days of instrument runtime and a few days of library-building. The aim of the Ion Analytics workflow is to automate the data processing aspect of library-building and once constructed employ GC/MS with either low- or high-resolution spectrometers to profile new samples.

Our approach allowed us to measure the differences in concentration between and among different quality green and roasted coffees, which should provide the ability to authenticate coffee by quality and/or region as we did for tea[11]. For example, coffee quality can be evaluated at various points along the supply chain, and higher quality designations will bring higher prices. Measuring cherry ripeness at harvest is one of the initial metrics of quality commonly used by produces around the world. Often, superior quality cherry with higher and consistent ripeness will be processed differently or simply tracked and kept separate, with a premium charged when the green coffee is sold for export. Further work will reveal how and by what means detailed chemical characteristics of coffee can be used to differentiate quality. In this study, we evaluated quality designations applied to the ripe coffee cherry by the Coopedota Cooperative in Tarrazú, Costa Rica, and found more than 100 chemical markers that differentiate the two highest quality (AAA and AA) coffees.

Round 2

Reviewer 1 Report

I appreciate the careful attention of the authors to my comments. The added subsection on the benefits of GC-GCMS enhances the text and the response regarding GC-GC vs GCxGC was also noted. The correction in figure caption 5 is likely a result of the final comment and acts to clarify the text. I also appreciated additional context relative to citations of the tea studies.